# Efficacy of two rounds of albendazole treatment on soil-transmitted helminths in schoolchildren, Yunnan Province, China

Darren J. Gray[1,2,12] ✉, Zunwei Du [3,12], Mary Lorraine Mationg[1,2], Yuesheng Li[1], Henglin Yang[3], Dongxu Wang[2], Eindra Aung[2,4,5], Franziska Angly Bieri[2], Suji O'Connor [2], Xinliu Yan [3], Fangwei Wu[3], Peter Steinmann[6,7], Kate Halton[8], Donald E. Stewart [9], Archie CA Clements[10], Donald P. McManus[1] & Gail M. Williams[11]

Mass drug administration (MDA) of albendazole to at-risk populations remains the primary strategy for controlling soil-transmitted helminths (STH). Despite its widely use, its efficacy varies among different STH species and remains sub-optimal, particularly in the treatment of *T. trichiura*. Currently, studies investigating the optimal dose and regimens for albendazole are lacking. A long-itudinal cohort study was conducted to assess the efficacy of two single-dose albendazole 400 mg treatments given four weeks apart targeting STH infections compared with just one single-dose albendazole 400 mg on 375 schoolchildren in Bulang Shan, Menghai county, Yunnan Province, China from October to December 2015. The first round of albendazole resulted in cure rates (CR) of 92.5%, 63.1% and 5.1%, and egg reduction rates (ERR) of 99.2%, 87.9% and 41.1% for *A. lumbricoides*, hookworms and *T. trichiura*, respectively. With the second round, efficacy remains high against *A. lumbricoides* (98.9% CR), is increased against hookworm (92.2% CR), and remains low against *T. trichiura* (6.3% CR). The second round increased the ERR to 99.6%, 99.8% and 74.1% for the same species, respectively. In this setting, albendazole is thus highly effective against *A. lumbricoides*, reasonably effective against hookworm, but has low efficacy against *T. trichiura* following two rounds of treatment.

Around 900 million people are estimated to be infected by soil-transmitted helminths (STHs), which is considered to be one of the leading causes of ill-health caused by neglected tropical diseases globally[1]. The three main STHs—*Ascaris lumbricoides*, *Trichuris* trichiura, and hookworms—infect 445 million, 360 million, and 173 million people, respectively[1]. STH infections occur more commonly in the world's poorest tropical communities[2] where access to basic living amenities like toilets and water is insufficient, if not lacking. STH thus is

[1]QIMR Berghofer Medical Research Institute, Brisbane, QLD, Australia. [2]National Centre for Epidemiology and Population Health, Australian National University, Canberra, ACT, Australia. [3]Yunnan Institute for Parasitic Diseases, Puer, China. [4]School of Clinical Medicine, University of New South Wales, Sydney, NSW, Australia. [5]Pain Management Research Institute, Kolling Institute, Northern Sydney Local Health District, University of Sydney, Sydney, NSW, Australia. [6]Swiss Tropical and Public Health Institute, Allschwil, Switzerland. [7]University of Basel, Basel, Switzerland. [8]School of Public Health and Social Work, Queensland University of Technology, Brisbane, QLD, Australia. [9]School of Medicine and Dentistry, Griffith University, Gold Coast, QLD, Australia. [10]Queens University Belfast, Belfast, Northern Ireland, UK. [11]School of Public Health, University of Queensland, Brisbane, QLD, Australia. [12]These authors contributed equally: Darren J. Gray, Zunwei Du. ✉e-mail: Darren.Gray@qimrb.edu.au

a tropical disease of poverty as it is strongly associated with impoverished rural living conditions[3].

STH infections negatively impact economic development[4] because they cause reduced learning achievements and impaired adult productivity due to anemia and malnutrition[5], thereby restricting the adult's capacity to earn due to being sick and adding cost for medical treatments. STHs also have negative effects on childhood health and development, which could manifest as anemia[6], malnutrition[7], and stunted growth[8].

In China, national prevalence estimates of *A. lumbricoides*, *T. trichiura*, and hookworm infections in 1995 were 47.0%, 18.8%, and 17.2%, respectively[9]. Government-led pilot programs centered on large-scale population-based deworming together with health education and improved access to water sources and sanitary facilities have been successful in reducing the overall prevalence of STH infection in China, reported as 2.4% (*A. lumbricoides* 0.8%, hookworm 1.4%, *T. trichiura* 0.5%) in 2016[10]. However, prevalence remains higher in China's poorer southwest regions. In 2016, Yunnan Province, located in southwestern China and bordering Southeast Asian countries (Myanmar, Vietnam, Laos), had the highest reported prevalence of all STH infections (13%) across the country[10].

The main control strategy for STH infections is periodic mass drug administration (MDA) with preventive chemotherapy for STH, using drugs recommended by the World Health Organization (WHO), most commonly benzimidazoles: albendazole (400 mg single dose) and mebendazole (500 mg single dose)[11]. However, while the safety of these drugs has long been established[12], their efficacy across STH species is inconsistent. A network meta-analysis of randomized controlled trials has indicated that single-dose albendazole has high efficacy for *A. lumbricoides* (95.7% cure rate (CR)) and hookworm (79.5% CR) infections, but its efficacy is unacceptably low for *T. trichiura* (30.7% CR)[13].

To date, the optimal dose and regimens (frequency of treatment/dosing schedule) for albendazole have not been determined and are insufficiently optimized for treating STH[11,12,14,15]. Additional studies comparing different doses and regimens are still needed to identify optimal treatment strategies for MDA applications. The aim of this study was to determine if two single-dose albendazole 400 mg treatments 4 weeks apart offered a meaningful clinical improvement in CR and egg reduction rate (ERR) for STH infections, particularly *T. trichiura*, compared to just one single-dose of albendazole 400 mg.

## Results

### Sociodemographic and baseline characteristics
Of the 453 schoolchildren aged 5–16 years old who were assessed for enrollment eligibility, 375 (82.8%) were identified with a least one type of STH species by Kato–Katz (KK) at baseline and were treated with albendazole in schools by the research team. Most of these schoolchildren were infected with *T. trichiura* (74%), followed by *A. lumbricoides* (53.9%) and hookworm (37.7%). Of those schoolchildren positive for at least one type of STH infection, 47.2% had infections with two different worm species, while 26.4% had infections with one species and another 26.4% with three species. Co-infection with *A. lumbricoides* and *T. trichiura* was the most prevalent (64.9%) dual co-infection observed at baseline.

As detailed in Fig. 1, individuals who were positive for at least one STH infection were enrolled as a fixed cohort and followed up to assess the drug efficacy at baseline and FU treatments. Approximately half (50.7%) of this cohort were female, more than half (55.7%) were above the age of nine, and the majority (82.4%) were from the Bulang ethnic minority group (Table 1). The final FU cohort was 81.1% ($n = 304$) of the 375 fixed cohort. This final FU cohort consists of individuals who have available stool examination results from at least one follow-up (FU1 or FU2) and with two rounds of albendazole 400 mg treatments given

4 weeks apart. The final FU had similar baseline characteristics with the fixed cohort as presented in Table 1.

### Parasitological cure rates
Table 2 shows the albendazole CR for the three STH species at two FUs. The first round of albendazole CR for *A. lumbricoides* was high, at 92.5%, and the second round increased the CR to 98.9%. The difference in CRs between the first round and the second round regimens was statistically significant ($p = 0.001$). The first round of albendazole CR for hookworm was 63.1%, and the second round increased the CR to 92.2%. The difference in CRs between the first round and the second round regimens was statistically significant ($p < 0.001$). The first round CR for *T. trichiura* was 5.1%. The second round of albendazole only slightly increased this CR to 6.3%, and the difference in CRs between the first round and second round regimens was not statistically significant ($p = 0.62$). The McNemar's test was employed to calculate the $p$ values.

### Egg reduction rates
Table 3 shows the ERR for the three parasites assessed at both FUs. The first round of albendazole was associated with a very high 99.2% ERR for *A. lumbricoides*. The second round of albendazole increased the ERR of *A. lumbricoides* to 99.6%.

The first round of albendazole was associated with an 87.9% ERR for hookworm. The second round increased the ERR to 99.8%. However, the first round of albendazole was associated with a markedly lower ERR for *T. trichiura* compared to the other two STH species, at only 41.4%. The second round increased the *T. trichiura* ERR to 74.1%.

### Infection intensity
Table 4 shows the distribution of infection intensity categories by species and FU. The near 100% ERR of the first and second rounds of albendazole for *A. lumbricoides* is reflected in the change in intensity category distribution, with heavy infections being completely eliminated at FU1 and light/moderate infections being nearly completely eliminated at FU2. The majority of hookworm infections at baseline were in the light category, and nearly all moderate/heavy infections were eliminated at FU1. At FU2, moderate/heavy infections were completely eliminated. At baseline, approximately 33% of participants had moderate/heavy *T. trichiura* infections. At FU1, this was reduced to approximately 21%, and at FU2, this figure was only 9.3%. Heavy infections were completely eliminated at FU2.

### Adverse events
The number of participants reporting adverse events is presented in Table 5. Overall, at any FU, the proportion of participants reporting any adverse event was 37.8%. At FU1 (after the first single-dose treatment), 337 participants were interviewed for symptoms. A total of 120 (35.6%) participants reported any mild symptom, such as feeling sick (16.9%), dizziness (11%), diarrhea (7.4), and vomiting (1.5%). At FU2 (after the second single-dose treatment), the proportion of reporting any adverse event has slightly increased to 40.6% with similar events including feeling sick (16.2%), dizziness (15.9%), diarrhea (8.9%), and vomiting (3.3%). There were no serious adverse events observed in this study. Additionally, participants did not declare any existing infections or co-morbidities, nor did the study team observe any co-morbid condition.

## Discussion
Preventive chemotherapy as the main strategy for STH morbidity control heavily relies on two benzimidazole drugs: albendazole and mebendazole[11,15]. These two drugs have been extensively used worldwide for more than 50 years; however, their efficacy varies against all three STH species, with *T. trichiura* still remaining as the main

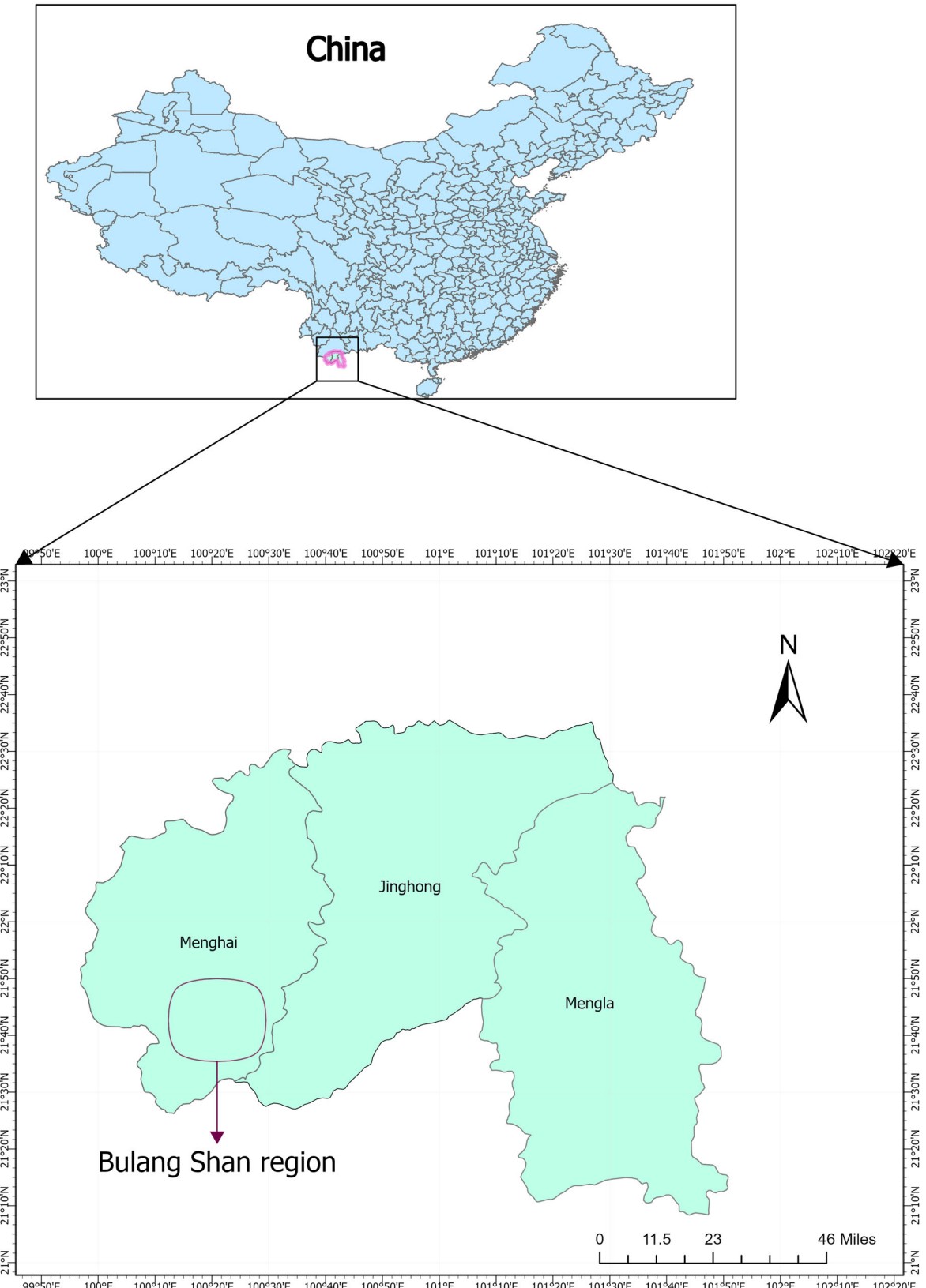

**Fig. 1 | Location of Bulang Shan Region in Menghai County, Yunnan Province, China.**

challenge[13,16]. Numerous studies have reported low efficacy of a single dose of 400 mg albendazole as recommended by the WHO against *T. trichiura*[13]. Despite the absence of effective alternative treatment against *T. trichiura*, only a few studies are available comparing the efficacy of different doses and regimens of albendazole[14]. This study aims to evaluate the effect of two rounds of single-dose albendazole 400 mg treatments administered 4 weeks apart on CR and ERR of STH infections, particularly *T. trichiura*.

As expected in this present study, single-dose albendazole resulted in a high level of efficacy in the treatment of *A. lumbricoides* (92.5% CR); however, the CRs were only 63.1% and 5.1% for hookworm and *T. trichiura* infections, respectively. These findings are comparable to the recent network meta-analysis of randomized controlled trials[13], demonstrating single-dose albendazole to be highly efficacious against *Ascaris* and moderately efficacious for hookworm infection, but not for *Trichuris*. Based on the recent meta-analysis, considering time interactions, a single dose of 400 mg albendazole has shown a limited efficacy against *T. trichiura*, with CRs having decreased from 38.6% in 1999 to 16.4% in 2015[13]. It is interesting to note that the reported efficacy of 6.3% for *T. trichiura* in this present study is by far the lowest observed following two doses compared to the average CR (30.7%) reported in the aforementioned network meta-analysis[13].

## Table 1 | Sociodemographic and baseline characteristics of the study participants

| Characteristics | Fixed cohort (*n* = 375) | | Follow-up cohort (*n* = 304) | |
|---|---|---|---|---|
| | No | % | No | % |
| **Sex** | | | | |
| Male | 176 | 46.9 | 147 | 48.4 |
| Female | 190 | 50.7 | 150 | 49.3 |
| Missing | 9 | 2.4 | 7 | 2.3 |
| **Age group** | | | | |
| Mean (SD)[a] | 10.12 (1.98) | | 10.14 (2.0) | |
| ≤7 years | 19 | 5.1 | 17 | 5.6 |
| 8–9 years | 139 | 37.1 | 109 | 35.9 |
| 10–11 years | 125 | 33.3 | 105 | 34.5 |
| >12 years | 84 | 22.4 | 67 | 22.0 |
| Missing | 8 | 2.1 | 6 | 2.0 |
| **Ethnicity** | | | | |
| Bulang | 309 | 82.4 | 262 | 86.2 |
| Lahu | 46 | 12.3 | 30 | 9.9 |
| Hani | 8 | 2.1 | 4 | 1.3 |
| Han | 3 | 0.8 | 1 | 0.3 |
| Yi | 1 | 0.3 | 1 | 0.3 |
| Missing | 8 | 2.1 | 6 | 2.0 |
| **Type of parasite co-infection** | | | | |
| Single parasite infection | 99 | 26.4 | 77 | 25.4 |
| Dual parasite infection | 177 | 47.2 | 146 | 48.0 |
| *Ascaris-Trichuris*[a] | 115 | 64.9 | 100 | 93.8 |
| *Trichuris-Hookworm*[a] | 55 | 31.1 | 41 | 28.1 |
| *Ascaris-Hookworm*[a] | 5 | 2.8 | 5 | 3.4 |
| Triple parasite infection | 99 | 26.4 | 81 | 26.6 |

[a]Dual parasite infection denominator *n* = 177 (fixed cohort) and *n* = 146 (follow-up cohort).

With regard to ERRs, single-dose albendazole resulted in an outstanding efficacy for *A. lumbricoides* with ERRs approaching 100%. Single-dose albendazole was also associated with an 87.9% ERR for hookworm. Compared to the other two STH species, a markedly lower ERR (41.4%) was observed for *T. trichiura*. For hookworm, the ERR reported from this study was higher than the ERR values reported from the meta-analysis of Moser et al.[13], while below the reference ERRs for the benzimidazoles published by WHO in 2013. The ERR for *T. trichiura* derived in this study was lower compared to the pooled ERR reported in the recent systematic review and meta-analysis[13] and the reference ERR by WHO[17]. It is noteworthy that treatment efficacy may vary across hookworm species[18], but we were unable to ascertain that here due to the use of KK and not molecular diagnostics.

With the second round of albendazole treatment, the observed CRs slightly increased against *A. lumbricoides* (98.9% CR), markedly increased against hookworm (92.2% CR), and remained low against *T. trichiura* (6.3% CR). The difference in CRs between first and second round regimens was statistically significant for *A. lumbricoides* and hookworm but not for *T. trichiura* in our study. The two repeated rounds (400 mg) did not offer improved efficacy (in terms of CR) against *T. trichiura*. In contrast to our study, Horton et al. in 2000 reported that increasing the single dosage and using repeated doses improves the efficacy of albendazole against *T. trichiura*[12]. However, in a recent trial investigating the feasibility of interrupting STH transmission using biannual albendazole community-based MDA (cMDA) compared to annual school-based MDA, repeated rounds of albendazole treatment led to interruption of hookworm (*N. americanus)* and any STH transmission, although there was no impact on *A. lumbricoides* and *T. trichiura* infections after 3 years of cMDA[19]. These differences could be attributed to decreasing efficacy over the last two decades and differing study design or geographic area.

The ERR for all three species after the administration of the second single-dose albendazole increased to 99.6%, 99.8% and 74.1% for *A. lumbricoides*, hookworm, and *T. trichiura*, respectively. These ERRs were above the ERR values for each species put forth by the meta-analysis of Moser et al.[13], and the reference ERRs by WHO[17]. The benefit of repeated doses is further shown in the significant reduction of moderate and heavy infections for *A. lumbricoides* and *T. trichiura*. The near 100% ERR for *A. lumbricoides* is reflected in the change in intensity category distribution, with heavy infections being completely eliminated at FU1 and light/moderate infections being nearly completely eliminated at FU2. At baseline, approximately 37.6% of participants had moderate/heavy *T. trichiura* infections. Moderate infection was reduced to approximately 21.4% at FU1 and 9.3% at FU2, and heavy infections were completely eliminated at FU2. Based on this result, from the public health perspective, it looks, therefore, that the two rounds of albendazole are effective in reducing infection intensity for *T. trichiura*, although the low CR found in this study is still worrying.

We acknowledge that this study lacks a comparison group as a limitation, which means no direct measure for comparative effectiveness. The current study was conducted to assess the added benefit of a

## Table 2 | Cure rates by species and follow-up

| Species | Time | In follow-up | Positive | Positive (%) (95% CI) | Cure rate % (95% CI) | *P* value FU2 vs FU1 |
|---|---|---|---|---|---|---|
| *A. lumbricoides* | FU1 | 200 | 15 | 7.5 (3.8, 11.2) | 92.5 (88.8, 96.2) | |
| | FU2 | 175 | 2 | 1.1 (0.0, 2.7) | 98.9 (97.3,100.4) | 0.001 |
| Hookworm | FU1 | 130 | 48 | 36.9 (28.5, 45.3) | 63.1 (54.7, 71.5) | |
| | FU2 | 116 | 9 | 7.8 (2.8, 12.7) | 92.2 (87.3, 97.2) | <0.001 |
| *T. trichiura* | FU1 | 272 | 258 | 94.9 (92.2, 97.5) | 5.1 (2.5, 7.8) | |
| | FU2 | 238 | 223 | 93.7 (90.6, 96.8) | 6.3 (3.2, 9.4) | 0.62 |

Two-sided *p* values were derived through McNemar's test.
*STH* soil-transmitted helminth, *FU1* follow-up 1, *FU2* follow-up 2.

**Table 3 | Egg reduction rates (ERR) by species and follow-up**

| Species | Time | Positive | Arithmetic mean EPG | Arithmetic mean EPG in positives | ERR (%) (95% CI) | ERR (%) (95% CI) in positives |
|---|---|---|---|---|---|---|
| A. lumbricoides | BL | 202 | 18,448 | 18,448 | - | - |
| | FU1 | 200 | 141 | 1885 | 99.2 (97.0, 99.8) | 91.6 (64.1, 98.0) |
| | FU2 | 175 | 67 | 5899 | 99.6 (97.4, 99.9) | 55.8 (−443, 96.4) |
| Hookworm | BL | 133 | 779 | 779 | - | - |
| | FU1 | 130 | 94 | 256 | 87.9 (48.0, 97.2) | 76.2 (−2.7, 94.5) |
| | FU2 | 116 | 2 | 23 | 99.8 (99.5, 99.9) | 97.5 (95.3, 98.6) |
| T. trichiura | BL | 277 | 1899 | 1899 | - | - |
| | FU1 | 272 | 1113 | 1174 | 41.4 (26.7, 53.2) | 41.6 (26.9, 53.4) |
| | FU2 | 238 | 493 | 526 | 74.1 (67.3, 79.4) | 74.8 (68.2, 80.0) |

*EPG egg counts per gram, BL baseline, FU1 follow-up 1, FU2 follow-up 2.*

**Table 4 | Infection intensity categories by species and follow-up**

| Species | Time | Total (N) | Not Infected | | Light | | Moderate | | Heavy | |
|---|---|---|---|---|---|---|---|---|---|---|
| | | | n | % (95% CI) | n | % (95% CI) | n | % (95% CI) | n | % (95% CI) |
| A. lumbricoides | BL | 202 | 0 | | 82 | 40.6 (33.8, 47.4) | 94 | 46.5(39.6, 53.5) | 26 | 12.9 (8.2, 17.5) |
| | FU1 | 200 | 185 | 92.5 (87.9, 95.7) | 13 | 6.5 (3.5, 10.8) | 2 | 1.0 (0.4, 2.4) | 0 | - |
| | FU2 | 175 | 173 | 98.9 (95.9, 99.8) | 1 | 0.5 (0.0, 1.6) | 1 | 0.5 (0, 1.6) | 0 | - |
| Hookworm | BL | 133 | 0 | - | 122 | 91.7 (87.0, 96.5) | 5 | 3.8 (0.5, 7.0) | 6 | 4.5 (0.9, 8.1) |
| | FU1 | 130 | 82 | 63.1 (54.1, 71.4) | 47 | 36.2 (27.9, 45.0) | 0 | - | 1 | 0.8 (0.0, 2.3) |
| | FU2 | 116 | 107 | 92.2 (85.7, 96.3) | 9 | 7.8 (3.6, 14.2) | 0 | - | 0 | |
| T. trichiura | BL | 277 | 0 | - | 173 | 62.5 (56.7, 68.2) | 91 | 32.9 (27.3, 38.4) | 13 | 4.7 (2.2, 7.2) |
| | FU1 | 272 | 14 | 5.2 (2.5, 7.7) | 195 | 71.7 (65.9, 76.9) | 59 | 21.6 (16.5, 26.2) | 4 | 1.5 (0.0, 2.9) |
| | FU2 | 238 | 15 | 6.3 (3.5, 10.2) | 199 | 83.6 (78.2, 88.1) | 24 | 10.13 (6.5, 14.6) | 0 | - |

*BL baseline, FU1 follow-up 1, FU2 follow-up 2.*

**Table 5 | Adverse effects**

| Event | Reported at FU1 (N = 337) | | Reported at FU2 (N = 217) | | Reported at any FU (N = 608) | |
|---|---|---|---|---|---|---|
| | n | % (95%CI) | n | % (95%CI) | n | % (95%CI) |
| Any AE | 120 | 35.6 (30.5, 40.7) | 110 | 40.6 (34.7, 46.5) | 230 | 37.8 (34.0, 41.7) |
| Felt Dizzy | 37 | 11.0 (7.6, 14.3) | 43 | 15.9 (11.5, 20.2) | 80 | 13.2 (10.5, 15.9) |
| Felt Sick | 57 | 16.9 (12.9, 20.9) | 44 | 16.2 (11.8, 20.7) | 101 | 16.6 (13.6, 19.6) |
| Diarrhea | 25 | 7.4 (4.6, 10.2) | 24 | 8.9 (5.5, 12.3) | 49 | 8.1 (5.9, 10.2) |
| Vomiting | 5 | 1.5 (0.2, 2.8) | 9 | 3.3 (1.2, 5.5) | 14 | 2.3 (1.1, 3.5) |
| Other | 2 | 0.6 (0.0, 1.4) | 2 | 0.7 (0.0, 1.8) | 4 | 0.7 (0.0, 1.3) |

*Participants could have reported several adverse events. 38, 104, no questionnaire at FU1 and FU2, respectively.*

second 400 mg dose of albendazole in addition to the standard regimen (single round 400 mg dose) in the context of a public health deworming program in a highly endemic setting, thus assignment of a control group (placebo) was unattainable and unethical to implement.

Although the KK technique is a widely used tool by control programs for assessing MDA effectiveness, it is not the most sensitive diagnostic test for STH infections[20–23]. The KK may fail to detect low-intensity infections, which could lead to underestimation of the actual prevalence. In the context of efficacy trials, this could result in falsely elevated CRs due to undetected residual low egg counts post treatment[22]. Despite these limitations, the current study considered the KK technique using multiple stools and slides as the appropriate procedure of choice since the polymerase chain reaction (PCR) technique was being optimized at the time of the study. In contrast to KK, PCR is semi-quantitative, which poses a limitation for measuring ERRs. It is well recognized that the sensitivity of KK improves with the use of

additional stool samples and slides[24], therefore, employing two stool samples and triplicate slides per sample in this study likely enhanced the accuracy for STH detection.

The goal of preventive chemotherapy against STH, however, is to eliminate moderate and heavy infection intensities, with the aim of reducing morbidity. Currently, there is an ongoing discussion on the validity of CR as an indicator for assessing drug efficacy and the impact of preventive chemotherapy[25]. It has been pointed out that CR is an efficient indicator of drug efficacy against bacterial diseases (for which it was originally developed) but less efficient for helminth infections because CR is influenced by the intensity of infection at baseline and by the sensitivity of the parasitological technique used. These concerns may be of less significance in relation to the present study, as the majority of the STH-positive individuals at baseline were in the low infection intensity category.

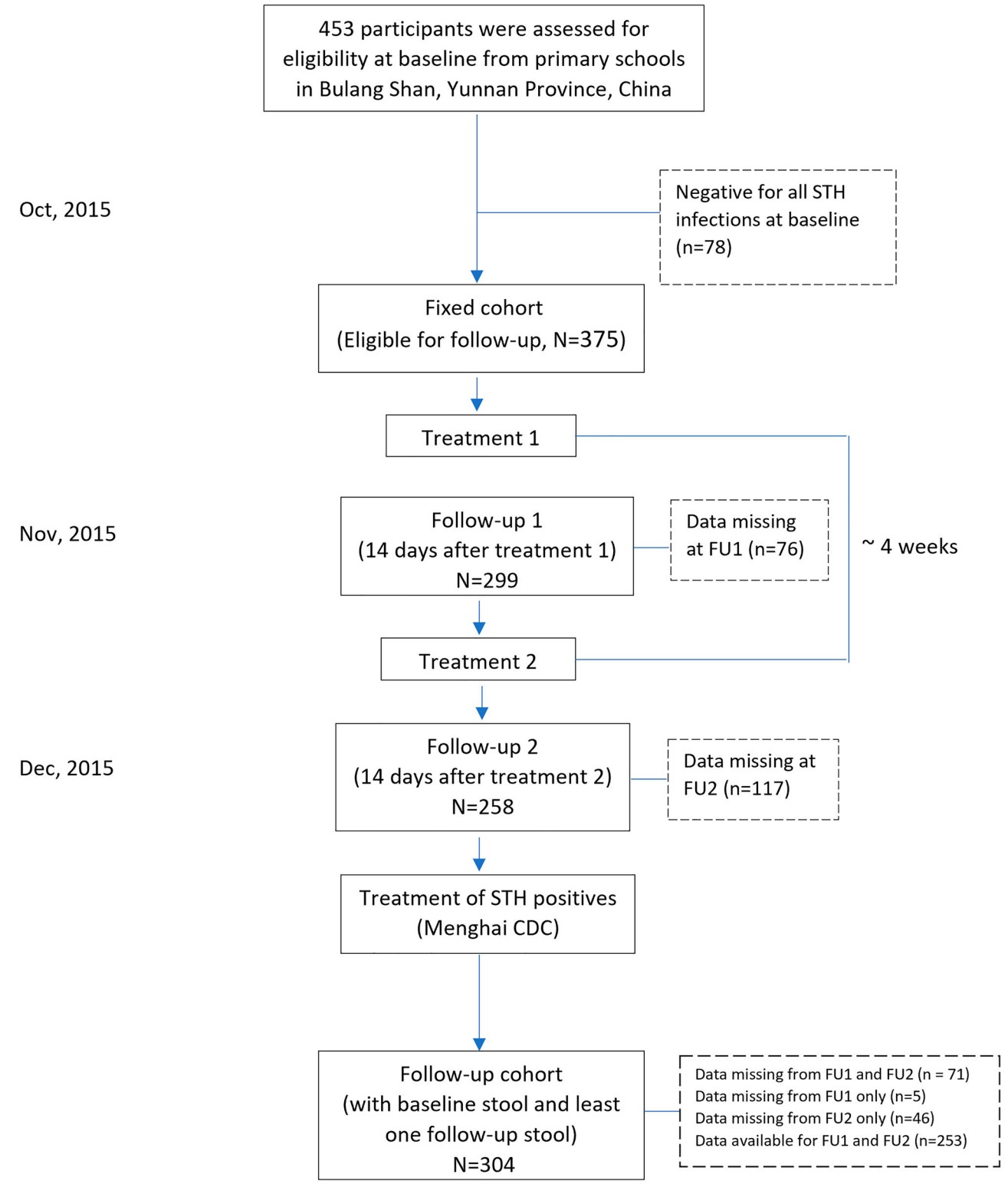

**Fig. 2** | Cohort profile.

The most notable finding of the study was the poor efficacy of albendazole against *T. trichiura*, resulting in only 6.3% efficacy even after the second round. This provides further evidence for the poor efficacy of albendazole against *T. trichiura* and reinforces the need for new STH treatments. There has been a growing body of research over the past 5 years advocating the use of ivermectin or oxantel pamoate as an alternative or supplementary treatment for STH infections. Both

drugs have shown high efficacy against *T. trichiura* in humans[13,26–31]. In 2017, the WHO included the co-administration of ivermectin with albendazole in their List of Essential Medicines to treat STH infections[28,32]. Moreover, recent studies reported that the use of moxidectin in combination with albendazole has shown enhanced efficacy against *T. trichiura*, and could be a potential alternative or complementary drug in STH control[30,33–35]. Further research should

explore the co-administration of albendazole with oxantel pamoate or moxidectin to improve current STH treatments. Additionally, drug administration alone is not enough to effectively control STH infections; an integrated approach combining improved water, sanitation, and health and hygiene education will be required for long-term sustainable control and elimination of STH infections.

## Methods

### Study design, setting, and population

This longitudinal cohort study was carried out in the Bulang Shan region, located in Menghai county, in Xishuangbanna autonomous prefecture, Yunnan Province, China, from October to December 2015, involving schoolchildren in grades 2–6 (aged 5–15 years) attending 22 primary schools. Figure 1 shows the map representing the location of the Bulang Shan region. The socio-economic status and household water, sanitation, and hygiene (WASH) conditions were homogeneous across the study area[36]. This region has been reported to have high levels of STH infections and rapid reinfections[36,37].

### Ethical considerations

The study was approved by the Yunnan Institute of Parasitic Diseases (China), Queensland Institute of Medical Research Human Research Ethics Committee (No. P1271), and the Australian National University Human Ethics Committee (No. 2014/356). We affirm that all study procedures contributing to this work complied with the ethical standards of these relevant committees. Before commencement of the study, informed consent was obtained from the parents or legal guardians of the study participants. For children aged >12 years, both informed consent from the parent or guardian and assent from the study participant were obtained. Identifying information, such as names, was kept confidential, and data was kept secure at all times.

### Study procedures

At baseline, we obtained one stool sample from all participating students. Three slides were prepared and examined microscopically (2–4 h post-collection to maximize hookworm diagnosis) using the KK thick-smear technique[38]. At follow-up 1 and 2, two stool samples were collected on separate days, and three slides per sample were prepared and examined using the same procedure.

A team of trained microscopists was organized to read the samples. Each microscopist independently read the samples assigned to them and was blinded to the results of other microscopists. The number of STH eggs was counted and recorded for each helminth species separately, and egg counts per gram (EPG) of feces were calculated based on the arithmetic mean of the three slide readings for the baseline sample and six slide readings for FU1 and FU2 samples, multiplied by a factor of 24 to determine infection intensities.

For quality control, 10% of the slides were rechecked by independent microscopists unaware of the initial results. Samples that were submitted more than 24 h after defecation were not accepted, and new containers were issued. Participants also completed questionnaires on demographic characteristics. All students who tested positive for at least one STH infection were recruited into the cohort (Fig. 2).

### Treatment delivery

All cohort members were treated with a dose of albendazole (400 mg oral tablet), and monitored for treatment compliance and acute side effects by the research team at each school. Four weeks after treatment, all cohort members were given their second dose of albendazole. The rationale behind the 4-week treatment interval was to allow immature worms in the circulation at the time of the first treatment to mature before administering the second treatment. Follow-up (FU) was conducted 14 days after each treatment. Treatment was recorded against the study participant's unique study ID. The treatment regimen employed was the standard of care at the time of the study, which has remained unchanged since then.

At each FU, all the assessments and quality-control measurements performed at baseline were repeated. In addition, participants completed questionnaires on post-treatment symptoms. In line with ethics procedures, any child who was still positive for STH after two albendazole treatments was referred to the Menghai Center for Disease Control and Prevention, Yunnan Province, for further medical attention (Fig. 2).

### Statistical analyses

The FU cohort was defined as those positive for any STH at baseline and who had stool examination results from at least one of the two rounds of FUs. Species-specific sub-cohorts of this FU cohort were defined as those positive for each of the respective species at baseline. CRs, defined as the proportion of children who were positive at baseline and became negative, were calculated for each FU, with 95% confidence intervals, within the relevant cohort. CRs for the two FUs were compared using McNemar's test. The arithmetic mean EPG, overall and for those positive only, and the percent ERR were calculated at the two FUs. Confidence intervals for the ERR were calculated using a formula based on the variance of the log (RR) and considering correlations between the baseline and FU. Intensity categories for *Ascaris lumbricoides*, hookworm, and *Trichiuris trichiura* were created based on WHO criteria as follows: light-intensity infections (EPG 0–5000, 0–2000, 0–1000, respectively); moderate-intensity infections (EPG 5001–50,000, 2001–4000, and 1001–10,000, respectively); and heavy-intensity infections (with EPG at or above the latter cut-off points, respectively)[39]. Occurrences of adverse effects were calculated from self-reports at FU1 (pertaining to baseline treatment) and FU2 (pertaining to FU1 treatment). The confidence interval for the combined FU1 and FU2 reports was calculated taking account of the correlation between the two reports.

Data were collected using paper-based forms and were double-entered into a customized password-protected Microsoft Access database. All data entered were saved offline, and backup paper duplicates were stored in a locked cabinet at Yunnan Institute of Parasitic Diseases. All data management and analyses used SAS (r) Proprietary Software 9.4 (TS1M7) [Copyright (c) 2016 by SAS Institute Inc., Cary, NC, USA], Licensed to Queensland Institute of Medical Research (QIMR)- Genetics and Population Health, Site 10008492.

### Reporting summary

Further information on research design is available in the Nature Portfolio Reporting Summary linked to this article.

## Data availability

Data supporting the findings of this study are not publicly available due to participant confidentiality. However, the data may be available from the corresponding author upon request. Access request will be responded to within 15 business days.

## Code availability

Custom code for data processing and analysis will be available from the corresponding author upon request. Access request will be responded to within 15 business days.

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

## Acknowledgements
This study was supported by the National Health and Medical Research Council Australia (Project Grant: 1046901) and the UBS Optimus Foundation, Switzerland.

## Author contributions
D.J.G., G.M.W. and D.P.M. conceived the study. D.G., D.M., G.W., Y.L., P.S., K.H. and A.C. obtained the funding. Z.D., D.W., X.Y., F.W., Y.L., H.Y., D.S., E.A., D.G. and F.B. undertook the fieldwork. M.L.M., G.W., S.O. and D.J.G. performed data analyses. All authors contributed to drafting and editing the manuscript.

## Competing interests
The authors declare no competing interests.
