## [Peer Review file · Nature Communications]

Efficacy of two rounds of albendazole treatment on soil-transmitted helminths in schoolchildren, Yunnan Province, China

Corresponding Author: Professor Darren Gray

Version 0:

Reviewer comments:

Reviewer #1

(Remarks to the Author)

This study aimed to evaluate the treatment efficacy of two rounds of albendazole administered two weeks apart. The treatment regimen was highly efficacious against *Ascaris*, moderately efficacious against hookworm, and demonstrated very low efficacy against *Trichuris*. As noted by the authors, the findings align with findings from previous studies and meta-analyses. The results also showed that a second round of treatment offered minimal benefit.

While the study was well-conducted, it had two key limitations. First, this was a longitudinal study and only 68% of children were successfully follow-up, thereby introducing potential bias. children were successfully followed up, potentially introducing bias. Data were collected only on age, sex, and ethnicity, meaning unmeasured exposure-related factors could have influenced the results.

Second, parasitological outcomes were assessed using repeat Kato-Katz smears. PCR is increasingly being used for STH diagnosis and provides higher sensitivity compared to traditional stool examination, especially for low-intensity infections. As such, the study may have under-estimated the true efficacy of treatment. These limitations are worth discussing.

Reviewer #2

(Remarks to the Author)

This longitudinal cohort study compares the effectiveness of two doses of albendazole (400 mg oral tablet), administered four weeks apart, with a single dose in treating soil-transmitted helminth infections among 375 schoolchildren (aged 5–15 years) in Yunnan Province, China. The effectiveness of the additional dose of albendazole was assessed through cure rate (CR) and egg reduction rate (ERR) for STH infections. Although highly effective against *Ascaris* and moderately effective against hookworm, two rounds of albendazole treatment had a 6.3% CR and 74.1% ERR against *Trichuris trichuria* infections. This study adds to the body of evidence demonstrating low effectiveness of albendazole in treating *Trichuris trichuria* infections.

I have a few comments for the authors to consider:

Major comments:

- (1) Please mention the rationale for the four-week interval in the treatment regimen of albendazole.
- (2) Were six Kato-Katz smears prepared from the single stool sample? If yes, how were the egg counts obtained if more than one smear was found to be positive? Also, what was the time interval between stool sample collection and testing? This is especially important for hookworm, as the eggs can degenerate over time. Moreover, in low-prevalence settings (as can be anticipated after two rounds of albendazole treatment) a single stool sample may result in false-negative findings due to day-to-day variation in egg excretion. This, in turn, could lead to an overestimation of both the CR and the ERR.
- (3) Did all the 304 children included in the follow-up cohort receive two doses of albendazole as per protocol?
- (4) The combined any-STH CR data is not particularly informative as it is influenced by the species-mix (in this study it is primarily driven by the high *Trichuris* prevalence). Please consider removing it from the manuscript.

(5) It will be interesting to see if and by how much the CR and ERR differed by infection intensity. Also, was the CR and ERR different in children with single vs. multiple infections?

(6) It is interesting to note that two doses of albendazole had a low CR (6.3%) but a relatively moderate ERR (74.1%) for *Trichuris* infections in this study? What could be the reason(s) for this observation? Would additional rounds of treatment with albendazole helped improve the CR in children with *Trichuris* infection?

Minor comments:

(1) A map of the study area will be helpful.

(2) Please consider adding a section on the study limitations.

(3) Was there any serious adverse event (SAE) reported in this study?

(4) Some of the 95% CIs presented in Tables 2, 4, and 5 (e.g., the 95% CI for the percentage of *Ascaris* infections in Table 2, or the percentage of light/moderate *Ascaris* infections in Table 4) appear to include negative values. I wonder if this is correct, as proportions cannot be negative. Could the authors please clarify?

Reviewer #3

(Remarks to the Author)

Author: Gray et al.

Title: Efficacy of two rounds of albendazole treatment on soil-transmitted helminths in school children, Yunnan Province, China

Journal: Nature Communication

Reference: NCOMMS-24-83683

General comment

The authors of the present work have made an impressive effort to collect data from a longitudinal cohort study and have generated a large body of data that will help to elucidate a large number of questions related to the elimination of STH. The manuscript is well written and structured. I have only a few comments for improvement.

Methods

- Did the study district was experiencing the MDA round before the startup of the current project? Is it possible to get the number of MDA rounds before your project? At least for your project areas?

- Treatment: "Once treated, SAC children participant had their finger marked with ink and the drug has been administrated by the teacher or the nurse? Otherwise this need some clarification

- Diagnostic: Kato-Katz was used and Six Kato-Katz thick smears were prepared per stool. Can you add some clarification about slide readers (i.e How many readers for the six smears? If several, what is the confidence rate? ...)

- Ethical consideration: please add the Yunnan Institute of Parasitic Diseases approval reference.

- Data Collection: I guess that the data were recorded electronically using Android phones with appropriated software (SurveyCTO or RedCap, or other). Could you add some detail please?

Results

- "A total of 375 individuals were identified with STH infection by Kato-Katz at baseline and were treated with a single dose of albendazole in schools or by the study team" Did the 375 individuals received a particular treatment in addition to the cMDA or SBD? If yes! This could be a source of bias.

- 230 AEs were reported in total during the study period. How many Adverse Effects classified as Serious (SAEs) have been reported? If none, it will be important to notify this. If participants have specific infection profile or any co-morbidity observed or declared? Good to mention for the readers

Discussion

Well written however a recent trial study about the feasibility of elimination STH transmission using albendazole need to be used to better discuss this work.

Minors

Line 153: The FU cohort as defined.... Please write: The FU cohort was defined....

Line 157: replace ... with 95% confidence intervals by at 95% confidence intervals

Line 211: replace a small benefit.... by lesser benefit....

Line 213: the phrase: "The second round increased the ERR to 99.8%" seem isolated. The second round increased ERR of which of the STH?

Line 333: replace STH infections long term.... By STH infections in a long term.....

Version 1:

Reviewer comments:

Reviewer #2

(Remarks to the Author)

I have now reviewed the revised manuscript and the author responses to the reviewers' comments. The revisions incorporated in response to the reviewers' feedback have substantially improved the clarity and overall presentation of the manuscript. I have a few additional (minor) comments for the authors to consider:

1. Introduction: A brief description of the Government-led STH prevention & control strategy in the study area will be helpful.
2. Results, Table 1: Please include a footnote that the denominator to calculate the percentages of *Ascaris-Trichuris*, *Trichuris*-Hookworm and *Ascaris*-Hookworm co-infection are those with dual parasite infection (n=177).
3. Results, Table 4: please provide the overall 'N', as well as the 'n' for each STH species. Also, although only the participants with at least one STH infection at baseline were included in the study, there will be a percentage of participants who were not infected by a specific STH species at baseline. Please modify the table to reflect this. At each time-point (BL, FU1, FU2) should the row percentages not add up to a total of 100%? For *T. trichuria*, the proportion of participants with moderate infection at FU2 is 9.3% as per table 2 but, in the text (lines 263-264), it is mentioned as 8.7% – please modify the text to reflect the correct value.
4. Table 5: The column 'N' for each time-point (FU1, FU2 & any FU) can be removed & the 'N' mentioned below the respective column header. Also, please add a footnote clarifying that participants could have reported more than one adverse event. Did the authors check for relatedness of the adverse event to the treatment? If yes, how many of the adverse events were considered as related to the treatment?
5. Discussion, lines 311-315: Could the low cure rates for *T. trichuria* observed in this study, as compared to the meta-analysis, be due to a relatively higher baseline prevalence of moderate-to-heavy *T. trichuria* infection among the study participants? This may also explain the higher ERR despite low CR at FU2, along-with a shift from moderate/heavy to light infections, considering the dose-dependent effect of albendazole on the intensity, but not on the prevalence of infection.
6. Figure 2 is not fully visible in the revised manuscript file.

No.	Reviewers' Comments	Authors' responses
1	Reviewer #1 This study aimed to evaluate the treatment efficacy of two rounds of albendazole administered two weeks apart. The treatment regimen was highly efficacious against Ascaris, moderately efficacious against hookworm, and demonstrated very low efficacy against Trichuris. As noted by the authors, the findings align with findings from previous studies and meta-analyses. The results also showed that a second round of treatment offered minimal benefit.	We thank you for your comments. Please find below our responses to each of your comments.
2	While the study was well-conducted, it had two key limitations. First, this was a longitudinal study and only 68% of children were successfully follow-up, thereby introducing potential bias.	The sample size for this study was estimated based on treatment efficacy of single dose albendazole (60%) and repeated dose albendazole (85%) on STH infections, $\alpha = 0.05$ and, $1-\beta=0.90$, a sample size of 128 STH positive children was required. Considering some children may miss the second treatment (20% attrition), thus a total of 153 STH positive children were targeted at the beginning of the study. To ensure the sample size can be achieved, the present study recruited 453 schoolchildren and screened for STH infection. Of these, 375 were found STH positive and enrolled for follow-up (fixed cohort). Of the 375 positive individuals at baseline, 304 (81%) had stool examination results from at least one of the two rounds of follow-up and this comprised our final follow-up cohort. The 71/375 positive individuals at baseline were excluded since they did not submit stool samples at any follow-up (FU1 or FU2). This exclusion is unlikely to result in a biased assessment of the effect of the treatment. As presented in Table 1, the final FU (n=304) had similar baseline characteristics with the fixed cohort (n=375).

3	Data were collected only on age, sex, and ethnicity, meaning unmeasured exposure-related factors could have influenced the results.	Aside from age, sex and ethnicity, we recognized that socio-economic status (SES) and school as well as household WASH conditions are potential exposure factors that could influence the study results. However, it is noteworthy that SES and WASH conditions were homogeneous across the study area. See reference 16. We have provided clarity on this in the study design section of the manuscript (see lines 96-98).
4	Second, parasitological outcomes were assessed using repeat Kato-Katz smears. PCR is increasingly being used for STH diagnosis and provides higher sensitivity compared to traditional stool examination, especially for low-intensity infections. As such, the study may have underestimated the true efficacy of treatment. These limitations are worth discussing.	Although, the KK technique being a widely used tool by control programs for assessing MDA effectiveness, it is not the most sensitive diagnostic test for STH infections (23-26). The KK may fail to detect low-intensity infections which could lead to underestimation of the actual prevalence. In the context of efficacy trials, this could result in falsely elevated CRs due to undetected residual low egg counts post treatment (25). Despite these limitations, the current study considered the KK technique using multiple stools and slides as the appropriate procedure of choice since polymerase chain reaction (PCR) technique was being optimised at the time of the study. In contrast to KK, PCR is semi-quantitative which poses a limitation for measuring egg reduction rates. It is well recognized that the sensitivity of KK improves with the use of additional stool samples and slides (27), therefore, employing two stool samples and triplicate slides per sample in this study likely enhanced the accuracy for STH detection. We have updated the manuscript to reflect our responses, please see lines 368-378.
Reviewer #2		
5	This longitudinal cohort study compares the effectiveness of two doses of albendazole (400 mg oral tablet), administered four weeks apart, with a single dose in treating soil-transmitted helminth infections among 375 schoolchildren (aged 5–15 years) in Yunnan Province, China. The effectiveness of the additional dose of albendazole was	Thank you for your comments. We have provided our responses to each of your comments below.

	assessed through cure rate (CR) and egg reduction rate (ERR) for STH infections. Although highly effective against Ascaris and moderately effective against hookworm, two rounds of albendazole treatment had a 6.3% CR and 74.1% ERR against Trichuris trichuria infections. This study adds to the body of evidence demonstrating low effectiveness of albendazole in treating Trichuris trichuria infections. I have a few comments for the authors to consider:	
6	Major comments: (1) Please mention the rationale for the four-week interval in the treatment regimen of albendazole.	The rationale behind the four-week treatment interval was to allow immature worms in the circulation at the time of the first treatment to mature before administering the second treatment. See lines 131-137
7	(2) Were six Kato-Katz smears prepared from the single stool sample? If yes, how were the egg counts obtained if more than one smear was found to be positive? Also, what was the time interval between stool sample collection and testing? This is especially important for hookworm, as the eggs can degenerate over time. Moreover, in low-prevalence settings (as can be anticipated after two rounds of albendazole treatment) a single stool sample may result in false-negative findings due to day-to-day variation in egg excretion. This, in turn, could lead to an overestimation of both the CR and the ERR.	At baseline, we obtained one stool sample from all participating students. Three slides were prepared and examined microscopically (2-4 hours post collection to maximize hookworm diagnosis) using the Kato-Katz (KK) thick-smear technique (18). At follow-up 1 and 2, two samples were collected and three slides per sample were prepared and examined using the same procedure. A team of trained microscopists was organized to read the samples. Each microscopist independently read the samples assigned to them and were blinded of the results of other microscopist. The number of STH eggs was counted and recorded for each helminth species separately and egg counts per gram (EPG) of faeces were calculated based on the arithmetic mean of the three slide readings for baseline sample and six slide readings for FU1 and FU2 samples multiplied by a factor of 24 to determine infection intensities. We have provided additional information on the manuscript, please see line 114-123. Please refer to our responses in item number 4. This limitation has been

		included in the discussion section of the manuscript, please see lines 368-378.
8	(3) Did all the 304 children included in the follow-up cohort receive two doses of albendazole as per protocol?	All 304 children included in the follow-up cohort received the two doses of treatment as per the study protocol. Please see section on treatment delivery, lines 130-137.
9	(4) The combined any-STH CR data is not particularly informative as it is influenced by the species-mix (in this study it is primarily driven by the high Trichuris prevalence). Please consider removing it from the manuscript.	We have excluded this in the manuscript.
10	(5) It will be interesting to see if and by how much the CR and ERR differed by infection intensity. Also, was the CR and ERR different in children with single vs. multiple infections?	Please refer to our result “infection intensity” section and Table 4. Here we described the distribution of infection intensity categories by species and discussed results in relation to CR and ERR (see lines 264-273). We did not specifically look at this as we were not powered for this sub-analysis
11	(6) It is interesting to note that two doses of albendazole had a low CR (6.3%) but a relatively moderate ERR (74.1%) for Trichuris infections in this study? What could be the reason(s) for this observation? Would additional rounds of treatment with albendazole helped improve the CR in children with Trichuris infection?	Yes, that is because albendazole does have an impact on Trichuris but not enough to cure. This could drive resistance though. An additional round could help but may not be practical for community-based interventions. For individual treatment, impact depends on compliance and re-infection.
12	Minor comments: (1) A map of the study area will be helpful.	A map of the study area has been included.
13	(2) Please consider adding a section on the study limitations.	The limitations of the study were included in the discussion section of the manuscript, please see lines 362-378.
14	(3) Was there any serious adverse event (SAE) reported in this study?	There was no serious adverse event observed in this study. This has been reflected in the manuscript, see lines 282-285.
15	(4) Some of the 95% CIs presented in Tables 2, 4, and 5 (e.g., the 95% CI for the percentage of Ascaris infections in Table 2, or the percentage of light/moderate Ascaris infections in Table 4) appear to include negative values. I wonder if this is correct, as proportions cannot be negative. Could the authors please clarify?	Values for confidence limits for proportions can fall outside the range 0 to 1 owing to the (usual) use of the normal approximation for the binomial distribution in the calculations. In the present case several lower limits fall just below zero. These have been set to zero, the implication being that the coverage of

		the 95% interval will be slightly less than the nominal value.
	Reviewer #3	
16	Author: Gray et al. Title: Efficacy of two rounds of albendazole treatment on soil-transmitted helminths in school children, Yunnan Province, China Journal: Nature Communication Reference: NCOMMS-24-83683 General comment The authors of the present work have made an impressive effort to collect data from a longitudinal cohort study and have generated a large body of data that will help to elucidate a large number of questions related to the elimination of STH. The manuscript is well written and structured. I have only a few comments for improvement.	Thank you for your comments. Please see our point-by-point responses below.
17	Methods - Did the study district was experiencing the MDA round before the start-up of the current project? Is it possible to get the number of MDA rounds before your project? At least for your project areas?	No regular/organized MDA was conducted before the start of the current project. Information on individual treatment is unavailable.
18	- Treatment: "Once treated, SAC children participant had their finger marked with ink and the drug has been administrated by the teacher or the nurse? Otherwise this need some clarification	The treatment was performed by the research team and recorded against the participants unique study ID The research team was also responsible for monitoring of treatment compliance and adverse events. Clarification has been made in the treatment delivery section. The result section (see lines 210-211) has been updated to provide further clarity.
19	- Diagnostic: Kato-Katz was used and Six Kato-Katz thick smears were prepared per stool. Can you add some clarification about slide readers (i.e How many readers for the six smears? If several, what is the confidence rate? ...)	Same as above. Please see authors' responses in item no. 4
20	- Ethical consideration: please add the Yunnan Institute of Parasitic Diseases approval reference.	The approval did not contain a reference number.
21	- Data Collection: I guess that the data were recorded electronically using Android phones with appropriated software	The data were collected on paper forms and double-entered into a customised password-protected Microsoft Access database. All data entered were saved

	(SurveyCTO or RedCap, or other). Could you add some detail please?	offline and back-up paper duplicates were stored in a locked cabinet at Yunnan Institute of Parasitic Diseases. All data management and analyses used SAS (r) Proprietary Software 9.4 (TS1M7) [Copyright (c) 2016 by SAS Institute Inc., Cary, NC, USA, Licensed to Queensland Institute of Medical Research (QIMR)-Genetics and Population Health, Site 10008492. This has been included in the manuscript, see lines 201-207.
22	Results - "A total of 375 individuals were identified with STH infection by Kato-Katz at baseline and were treated with a single dose of albendazole in schools or by the study team"	The treatment was done by the research team. The research team was responsible for monitoring of treatment compliance and adverse events. Clarification has been made in the treatment delivery section. The result section (see lines 211-212) has been updated to provide further clarity.
23	Did the 375 individuals received a particular treatment in addition to the cMDA or SBD? If yes! This could be a source of bias.	These 375 were found positive for STH (at eligibility assessment) and were suitable for follow-up. They were only given the first dose (treatment 1) and second dose (treatment 2) only with four weeks interval as per the study protocol. No additional treatments were given.
24	- 230 AEs were reported in total during the study period. How many Adverse Effects classified as Serious (SAEs) have been reported? If none, it will be important to notify this. If participants have specific infection profile or any co-morbidity observed or declared? Good to mention for the readers	There were no serious adverse events reported. This has been reflected in the manuscript, see lines 282-285. Additionally, participants did not declare any existing infections or comorbidities nor did the study team observe any co-morbid conditions. This information has been included as well, see lines 282-285
25	Discussion Well written however a recent trial study about the feasibility of elimination STH transmission using albendazole need to be used to better discuss this work.	We have included this reference in the discussion (see lines 342-346).
	Minors	
26	Line 153: The FU cohort as defined.... Please write: The FU cohort was defined....	This has been updated in the manuscript, see line 184.
27	Line 157: replace ... with 95% confidence intervals by at 95% confidence intervals	This has been updated in the manuscript, see line 188.

28	Line 211: replace a small benefit.... by lesser benefit....	We have excluded this sentence in the manuscript in relation to comment in item 9.
29	Line 213: the phrase: "The second round increased the ERR to 99.8%" seem isolated. The second round increased ERR of which of the STH?	This statement refers to A. lumbricoides species. This has been updated in the manuscript, see line 255.
30	Line 333: replace STH infections long term.... By STH infections in a long term.....	We have improved this sentence in manuscript, see lines 401 to 403.

No.	Reviewers' Comments	Authors' responses
	Reviewer #2	
	Reviewer #2 (Remarks to the Author): I have now reviewed the revised manuscript and the author responses to the reviewers' comments. The revisions incorporated in response to the reviewers' feedback have substantially improved the clarity and overall presentation of the manuscript. I have a few additional (minor) comments for the authors to consider:	We thank you for your comments. Please find below our responses to each of your comments.
1	Introduction: A brief description of the Government-led STH prevention & control strategy in the study area will be helpful.	As noted to the previous response to the reviewer's comments, no regular or organized mass drug administration (MDA) had been conducted in the study area prior to the commencement of the current project. Although, we have included a brief description in the introduction regarding the Government-led STH control program conducted in China more broadly. We confirm that these initiatives had not been applied to the specific study area during the study period.
2	Results, Table 1: Please include a footnote that the denominator to calculate the percentages of Ascaris-Trichuris, Trichuris-Hookworm and Ascaris-Hookworm co-infection are those with dual parasite infection (n=177).	This has been updated in the manuscript
3	Results, Table 4: please provide the overall 'N', as well as the 'n' for each STH species. Also, although only the participants with at least one STH infection at baseline were included in the study, there will be a percentage of participants who were not infected by a specific STH species at baseline. Please modify the table to reflect this. At each time-point (BL, FU1, FU2) should the row percentages not add up to a total of 100%? For T. trichuria, the proportion of participants with moderate infection at FU2 is 9.3% as per table 2 but, in the text (lines 263-264), it is mentioned as 8.7% – please modify the text to reflect the correct value.	Thank you for your comment. We have updated the table to reflect the overall sample size (N) and have corrected the percentages and the CI accordingly. This has been updated in the manuscript.

4	Table 5: The column ‘N’ for each time-point (FU1, FU2 & any FU) can be removed & the ‘N’ mentioned below the respective column header. Also, please add a footnote clarifying that participants could have reported more than one adverse event. Did the authors check for relatedness of the adverse event to the treatment? If yes, how many of the adverse events were considered as related to the treatment?	This has been updated in the manuscript. As outlined in line 278-287, participants were interviewed following treatment, indicating that the adverse events described are directly related to the treatment experience.
5	Discussion, lines 311-315: Could the low cure rates for T. trichuria observed in this study, as compared to the meta-analysis, be due to a relatively higher baseline prevalence of moderate-to-heavy T. trichuria infection among the study participants? This may also explain the higher ERR despite low CR at FU2, along-with a shift from moderate/heavy to light infections, considering the dose-dependent effect of albendazole on the intensity, but not on the prevalence of infection.	This may not be the case for the current study, given that the majority (62.5%) of study participants with baseline Trichuris infection were classified as having light intensity infection.
6	. Figure 2 is not fully visible in the revised manuscript file.	The figure has been updated in the revised manuscript.